# Genetic Deficiency of the Histamine H_4_-Receptor Reduces Experimental Colorectal Carcinogenesis in Mice

**DOI:** 10.3390/cancers12040912

**Published:** 2020-04-08

**Authors:** Bastian Schirmer, Tamina Rother, Inga Bruesch, Andre Bleich, Christopher Werlein, Danny Jonigk, Roland Seifert, Detlef Neumann

**Affiliations:** 1Institute of Pharmacology, Hannover Medical School, 30625 Hannover, Germany; 2Institute of Transfusion Medicine, Hannover Medical School, 30625 Hannover, Germany; 3Institute for Laboratory Animal Science, Hannover Medical School, 30625 Hannover, Germany; 4Institute of Pathology and German Center of Lung Research (DZL), Partner site BREATH, Hannover Medical School, 30625 Hannover, Germany

**Keywords:** histamine, H_4_ receptor, colorectal cancer, mouse model, cyclooxygenase 2

## Abstract

Colorectal cancer (CRC), a severe complication of inflammatory bowel diseases, is a common type of cancer and accounts for high mortality. CRC can be modeled in mice by application of the tumor promoter, azoxymethane (AOM), in combination with dextran sodium sulfate (DSS), which are able to induce colitis-like manifestations. Active colitis correlates with high mucosal concentrations of histamine, which, together with the histamine receptor subtype 4 (H_4_R), provide a pro-inflammatory function in a mouse colitis model. Here, we analyzed whether H_4_R is involved in the pathogenesis of AOM/DSS-induced CRC in mice. As compared to wild type (WT) mice, AOM/DSS-treated mice lacking H_4_R expression (TM) demonstrate ameliorated signs of CRC, i.e., significantly reduced loss of body weight, stiffer stool consistency, and less severe perianal bleeding. Importantly, numbers and diameters of tumors and the degree of colonic inflammation are dramatically reduced in TM mice as compared to WT mice. This is concomitant with a reduced colonic inflammatory response involving expression of cyclooxygenase 2 and the production of C-X-C motif chemokine ligand 1 (CXCL1) and CXCL2. We conclude that H_4_R is involved in the tumorigenesis of chemically-induced CRC in mice via cyclooxygenase 2 expression and, probably, CXCL1 and CXCL2 as effector molecules.

## 1. Introduction

The biogenic amine histamine (2-(4-imidazolyl)-ethylamine) is a profound mediator of inflammation [1]. It is recognized by the respective target cell through histamine-specific G protein-coupled receptors, which are subdivided into four subtypes: histamine H_1_-receptor (H_1_R), H_2_R, H_3_R and H_4_R. Out of these, the H_4_R is the latest one identified, due to its homology (37%) to the H_3_R [2,3,4,5,6]. H_4_R is predominantly expressed in immune cells [7,8,9,10,11] and its activation affects inflammatory and immune reactions via G_i_-mediated pathways [10,12,13,14]. 

Inflammatory bowel diseases (IBD), including the most prevalent manifestations, ulcerative colitis and Crohn’s disease, are idiopathic, chronic-recurring disorders of the gut. They severely affect the quality of patients’ lives and eventually limit their life expectancy through complications like extra-intestinal manifestations and colorectal cancer (CRC) [15,16]. CRC is the third most common type of cancer in men and accounts for high mortality [17,18]. In addition to colitis-associated CRC, CRC appears sporadic and hereditary [19,20]. However, although inflammation rarely precedes sporadic or hereditary CRC, anti-inflammatory drugs are effective in preventing or delaying the diseases. Thus, inflammatory reactions seem to be also involved in tumorigenesis of sporadic and hereditary cases [21,22].

Dextran sodium sulfate (DSS)-induced colitis is a widely used model for ulcerative colitis. The treatment of animals with the tumor promoter azoxymethane (AOM) in combination with DSS serves as a model for CRC (AOM/DSS model). DSS is thought to induce injury of colon epithelial cells, leading to impairment of the epithelial barrier and bacterial infiltration into the colonic mucosa [23]. The subsequent local inflammatory response is dominated by innate immune mechanisms [24,25]. Neutrophilic granulocytes are massively recruited to the lamina propria [26]. Other immune cells involved in colitic inflammation are mast cells, basophils, and eosinophils [27]. 

The increased number of mucosal mast cells able to release histamine has been shown to correlate with active colitis [28], as does the enhanced mucosal concentration of histamine [29]. A pro-inflammatory function of histamine and the H_4_R in DSS-induced colitis in mice has been demonstrated in previous studies [30,31,32,33]. Thus, in the present study, we aimed at analyzing whether the H_4_R bears a functional role in the pathogenesis of chemically induced CRC employing the AOM/DSS model in mice.

## 2. Results

### 2.1. Clinical and Macroscopic Evaluation

Upon induction of CRC by AOM/DSS treatment (Figure 1a), the mice developed a transiently increasing and decreasing activity of intestinal disease (Figure 1b). Increases of disease activity coincided with DSS feeding, while decreases occurred during DSS-free periods. Application of the tumor promoter, AOM, at day 0 of the treatment schedule did not induce any obvious disease activity until the start of DSS treatment. Thus, control mice, which for the entire schedule were treated with AOM only and fed with water not supplemented, did not develop any significant signs of disease activity (Figure 1b). Remarkably, TM CRC mice demonstrated a significant lower disease activity as compared to WT CRC mice (Figure 1b).

Analyzing the colon specimen of these mice at the end of the treatment schedule revealed dramatic differences between WT CRC and TM CRC mice (Figure 2a). While WT CRC mice developed a high number of large tumors at the distal part of their colons, tumors in TM CRC mice were significantly lower in number and diameter (Figure 2b). Control mice treated only with AOM developed no colon tumors (Figure 2a,b).

### 2.2. Microscopic Evaluation of Colon Tissues

The colonic mucosa of AOM-only treated control mice demonstrated a normal histological appearance, showing no signs of tumor growth (Figure 3a) with a low inflammation score (Figure 3b). In contrast, histological colon specimens from WT CRC mice exhibited marked mucosal inflammation and tumor-adjacent mucosal hyperplasia (Figure 3a), resulting in an increased inflammation score compared to the specimen of the control mice (Figure 3b). These parameters appeared to be less pronounced in TM CRC mice as compared to WT CRC mice (Figure 3b), although the difference was not statistically significant. The inflammatory reaction was—in both WT CRC mice and TM CRC mice—pronounced in mucosal areas adjacent to the tumor, with areas far from the tumor being less affected. In addition, tumors from TM CRC mice showed a more sessile morphology, while WT CRC mice revealed a predominantly polypoid morphology. The tumor-adjacent inflammatory infiltrate consisted mainly of CD3^+^ lymphocytes and plasma cells (Appendix A). In direct vicinity to the infiltrate, neutrophilic granulocytes could also be found. The tumors themselves showed a broad tubular architecture with only scarce areas of cribriform or solid growth patterns with marked nuclear pleomorphy and atypia, as well as an increased mitotic activity. None of the tumors infiltrated the tunica muscularis propria, with possible infiltrations of the submucosa. Thus, carcinogenesis in our model is still at an early stage, which is most likely due to the experimental schedule that may be too short to allow full carcinoma development. 

### 2.3. CXCL1 and CXCL2 Production

In colon homogenates and sera from control mice, C-X-C motif chemokine ligand 1 (CXCL1 and CXCL2 were barely detectable, while significant concentrations could be measured in homogenates and sera from WT CRC and TM CRC mice (Figure 4a–d). These concentrations, however, did not significantly differ between WT CRC and TM CRC mice. 

### 2.4. COX 2 and NOS 2 Expression

mRNA encoding cyclooxygenase 2 (COX 2), which is expressed in around 80% of human CRC, was detectable in colon specimen of control mice treated with AOM only, independent of the genotype. After CRC induction, the amount of COX 2 mRNA was significantly increased in WT CRC mice, while in TM CRC mice it remained at the level of control mice (Figure 5a). Thus, a significant difference in COX 2 mRNA accumulation was detected between colon samples of WT CRC and TM CRC mice. mRNA encoding the inducible nitric oxide synthase (nitric oxide synthase 2; NOS 2), which was reported to demonstrate an up-regulated expression in AOM/DSS-induced dysplasia in mice, in our hands, was detectable in all groups of samples but without significant differences regarding treatment or genotype (Figure 5b). 

## 3. Discussion

The present study aimed at analyzing whether or not histamine and its receptor H_4_R are involved in the pathogenesis of CRC, modelled by AOM/DSS treatment of mice, as already suggested by Tanaka et al. [34]. To this end, development of CRC was chemically induced in WT and TM mice, and symptoms as well as parameters indicative of intestinal disease were compared. The change in clinical symptoms analyzed in this study and reported as DAI (sum of weight loss, perianal bleeding, and reduced stool consistency) indicates colonic derangements and is significantly higher on the feeding of colitogenic DSS. Importantly, disease activity is reduced due to the absence of H_4_R expression, and thus confirms already reported data on the involvement of the H_4_R in experimental acute colitis in mice [30,33]. Moreover, it assigns H_4_R a function to a chronic model of colitis, beyond its recognized function to acute colitis models [30,33]. Certainly, the statistical significance of the difference between the DAI of WT CRC and TM CRC groups is limited, which is most probably due to the fact that the data of this small proof-of-concept study are based on only a very limited number of mice per group. This limitation of this study may also account for the poor or lacking significance of several other parameters that are discussed below. The rather low number of animals, however, was sufficient to reveal a significant difference when looking at the tumor burden, the primary goal of this study. 

Tumor development became obvious upon brief inspection of the colon specimen. We could confirm that cancerization was strictly associated to DSS-induced colitis, since control mice treated with only the tumor promoter, AOM, did not develop any colonic tumor. Both macroscopic and microscopic evaluations reveal that the absence of H_4_R reduced inflammation, leading to the reduced number and diameter of tumors. This observation is highly similar to findings observed in an experimental model of breast cancer [35]. 

Based on our findings, two hypothetical mechanisms may be the underlying cause: (1) lack of H_4_R expression reduces colitis as discussed above and, subsequently, dampens inflammatory-driven, colitis-dependent tumorigenesis, or (2) lack of H_4_R expression abolishes a tumor cell proliferation-promoting effect of histamine-induced H_4_R signaling. Of course, both mechanisms in combination may also explain the observed effect. 

Undoubtedly, histamine and its receptor H_4_R exert a pro-inflammatory function in DSS-induced colitis in mice [30,32,33], rendering hypothesis (1) feasible. This is supported by the slightly-enhanced production of CXCL1 and CXCL2 in AOM/DSS-treated mice as compared to AOM-only-treated mice. This rather modest increase is probably a consequence of the experiment’s timing (Figure 1a). If samples for expression/production analyses were generated immediately at the end of a cycle of DSS feeding, a significant increase in CXCL1 and CXCL2 expression was detected (unpublished observation), while in the present study, samples were generated roughly two weeks after the last cycle of DSS feeding. Thus, at this time point in the experimental protocol, expression of CXCL1 and CXCL2 is already declining.

On the other hand, CXCL1 [36] and CXCL2 [37], that are produced upon appropriate stimulation by a variety of cell types including monocytes/macrophages, neutrophils, epithelial cells and also tumor cells, have functions beyond attracting neutrophil granulocytes to the site of infection or damage. In colon cancer, they are associated with angiogenesis and the transition from dysplasia to carcinoma [38,39,40], indicating a possible mechanistic connection between inflammation and tumorigenesis. The expression of CXCL1 and CXCL2 in tumors is regulated, besides others, by prostaglandin E_2_ (PGE_2_) [38]. PGE_2_ is a lipid mediator that is generated by means of COX 2, a diagnostic marker for CRC [41]. The expression of COX 2, which is, in contrast to that of NOS 2, upregulated due to the AOM/DSS treatment, strictly depends on the presence of the H_4_R. Thus, H_4_R signaling in epithelial cells may induce COX 2 expression [42], leading to an increased production of PGE2, which, subsequently, induces the production of CXCL1 [43] and CXCL2, resulting in tumor progression. Supporting evidence comes from experiments showing that enhanced COX 2/PGE_2_ levels mediate the histamine-induced tumor vascularization and proliferation [42]. 

Myeloid-derived suppressor cells (MDSC) expressing CXCR2 can be recruited to the developing tumor by CXCL1 and CXCL2 [38], possibly induced by PGE_2_ [44]. MDSC are essential for tumor development, since they dampen the immune reaction against the tumor, e.g., by inhibition of CD8^+^ T cell activity [38]. MDSC express the histamine receptors H_1_R, H_2_R, and H_3_R [45]. Expression of H_4_R was not detected [45], while the H_4_R was readily detected in CRC-containing tissue [46]. Thus, a direct effect of histamine via H_4_R on MDSC can be excluded, suggesting that histamine via the H_4_R may promote tumorigenesis by indirectly enhancing the MDSC’s activity via epithelial cell-produced CXCL1 and CXCL2, or by a different mechanism not involving MDSC. Nevertheless, these hypotheses are in contrast to publications providing evidence for a MDSC-mediated CRC suppressing effect of histamine [47,48,49,50]. These studies, however, did not analyze the histamine receptor subtypes involved. Other publications indicate that histamine promotes the MDSC activity via H_1_R and H_2_R [45]. The overall role and effect of histamine on CRC development is discussed highly controversially, ranging from a proposed MDSC-mediated CRC suppression [46,47,48,49] to histamine-driven promotion of MDSC activity via H_1_R and H_2_R, which would result in hampered tumor progression [44]. Therefore, the CRC-promoting effect of the H_4_R we report here most probably is not mediated via an enhanced activity of MDSC. 

In summary, we have provided evidence that the presence of the H_4_R, most likely in epithelial cells [51], is necessary for the tumorigenesis in a mouse model of chemically-induced colorectal carcinoma. The mechanisms at the heart of this observation involve COX 2 expression and probably the production of the pro-inflammatory mediators CXCL1 and CXCL2. Further details, however, still have to be explored.

## 4. Materials and Methods 

### 4.1. Materials

If not stated otherwise, all chemicals were obtained from Sigma-Aldrich (Munich, Germany).

### 4.2. Animals

BALB/cJRj (WT) mice were purchased from Janvier Labs. Mice with deletion of functional H_4_R expression due to a targeted mutation of the H_4_R gene (TM; strain: C.129HrH_4_^tm1Lex^), generated by Lexicon Genetics (Woodlands, TX, USA) and described by Hofstra et al. [11], were backcrossed for more than 10 generations onto the BALB/cJRj strain. All animals were bred and maintained at the central animal facility of Hannover Medical School in a standardized environment (temperature: 21 °C +/− 1 °C; 14/10-h day/night cycle). They had access to standard diet (Altromin 1310, Altromin special diet, Lage, Germany) and drinking water ad libitum. The hygienic status of the mice was determined routinely according to the FELASA-guidelines to ensure absence of mouse pathogens. For the experiments, female mice of 10–20 weeks of age, randomly assigned to the experimental groups, were used. 

All applicable international, national, and/or institutional guidelines for the care and use of animals were followed. The study was conducted in accordance with the German law for animal protection (TierSchG) and with the European Directive 2010/63/EU. All experiments were approved by the Local Institutional Animal Care and Research Advisory Committee and permitted by the local government (AZ 33.12-42502-04-16/2197; July 29, 2016). 

### 4.3. Induction of CRC by AOM/DSS and Animal Dissection

For induction of CRC, mice were injected intraperitoneally once with AOM (10 mg/kg body weight) at day 0 and then fed for three cycles with water charged with 2.0% (w/v) DSS as depicted in Figure 1a. The first and second DSS cycle lasted for 7 days. The third cycle was terminated already after 5 days, since otherwise some mice probably would have had to be excluded from the experiment due to severely impaired health conditions. In the periods between the DSS cycles the mice received pure drinking water. AOM-treated but water-only fed mice served as control. Mice were inspected daily as detailed in Section 2.4. On day 68 the animals were euthanized with carbon dioxide insufflation and subsequent heart puncture to draw blood for sera preparation using Serum Gel Z/1.1 tubes (Sarstedt, Nümbrecht, Germany). Colon specimen were resected, washed with PBS to remove remaining feces, opened longitudinally, and photographically documented. Afterwards, colon tissues were divided longitudinally, and one part was processed for histological examination (see below), while the other part was divided into tumor-bearing and non-tumor-bearing sections and stored in RNA later (Thermo Fisher Scientific, Waltham, MA, USA).

### 4.4. Evaluation of Disease Activity

Mice were examined at 24-hour intervals using a common clinical score and a disease activity index (DAI, adopted from Alex et al. [52]) ranging from 0 to 12 was employed. The DAI was based on total body weight loss (0: no weight loss, 1: ≤5%, 2: ≤10%, 3: ≤15%, 4: >15%), stool consistency (0: normal, 2: soft, 4: diarrhea) and perianal bleeding (0: no bleeding, 2: little bleeding, 4: massive bleeding).

### 4.5. Histology

The respective parts of the colon tissues were fixed in 4% (v/v) formaldehyde (Merck, Darmstadt, Germany), embedded in paraffin, sliced, and stained with hematoxylin/eosin (H/E). Stained tissue slices were analyzed in a blinded fashion by two pathologists. A histological severity score was calculated by evaluating the severity (leukocyte density and area of affected lamina propria: 0: normal, 1: minimal (<10%), 2: mild (10–25%), 3: moderate (26–50%), 4: marked (>51%)) and the extent (expansion of leukocyte infiltration: 0: normal, 1: mucosal, 2: submucosal, 3: transmural) of the inflammatory cell infiltrate, mucosal hyperplasia (increase in epithelial cell numbers in longitudinal crypts: 0: normal, 1: minimal <25%, 2: mild 25–35%, 3: moderate 36–50%, 4: marked > 51%) and goblet cell loss (0: normal, 1: minimal: < 20%, 2: mild: 21–35%, 3: moderate: 36–50%, 4: marked >50%). The maximum theoretical score sums up to 15 points [53]. Further, the presence or absence of cancer as well as predominantly polypoid tumors was evaluated. Reported are the values obtained by the analyses of tumor-proximal areas. 

### 4.6. Soluble Mediator Protein Quantification

Portions (~30 mg) of the RNA later-stored colon tissues were lysed in RIPA buffer containing proteases by sonification and 30 minutes of rotation at 4 °C. Aggregates were removed by centrifugation (20 min, 10,000× *g*, 4 °C) and protein concentration in the supernatants was measured using the Pierce BCA protein assay (Thermo Fisher Scientific). Quantification of cytokine concentrations in the supernatants was carried out using the Magnetic Luminex Screening Assay (R&D Systems, Minneapolis, MN, USA). The kit quantified the cytokines KC/CXCL1, MIP-2/CXCL2, MCP-1/CCL2, IL-12p70, IL-1β, IL-5, IL-10, IFNγ, TNF, IL-6, IL-17A, and IL-23p19. 

### 4.7. Soluble Mediator mRNA Quantification

Portions (~30 mg) of the RNA later-stored tissues were homogenized using the FastPrep-24 device (MP Biochemicals, Irvine, CA, USA). RNA was extracted from the homogenates using the Nucleospin RNA II kit (Macherey-Nagel, Düren, Germany) essentially according to the manufacturer’s instructions. One µg RNA of each sample was reversely transcribed for 30 min at 50 °C into cDNA by means of Maxima Reverse Transcriptase (Thermo Fisher Scientific). Target gene-specific sequences were quantified proportionately to glycerin aldehyde-3-phosphate dehydrogenase (GAPDH) by real-time PCR using TaqMan Gene Expression Assays (ABI; Thermo Fisher Scientific). The reported relative gene expression data were calculated by relating the obtained qPCR data of corresponding tumor bearing and not tumor bearing sections by the ΔΔC_T_ method. 

### 4.8. Statistical Analysis

Data are represented as arithmetic means ± SD for each parameter. Statistical analyses were performed with GraphPad Prism version 6.07 (GraphPad Software, La Jolla, CA, USA), using ANOVA with Sidak’s post hoc test.

## 5. Conclusions

We conclude that histamine via the H_4_R promotes tumorigenesis in a mouse model of CRC. Whether this provides new therapeutic targets has to be evaluated in the future.

## Figures and Tables

**Figure 1 cancers-12-00912-f001:**
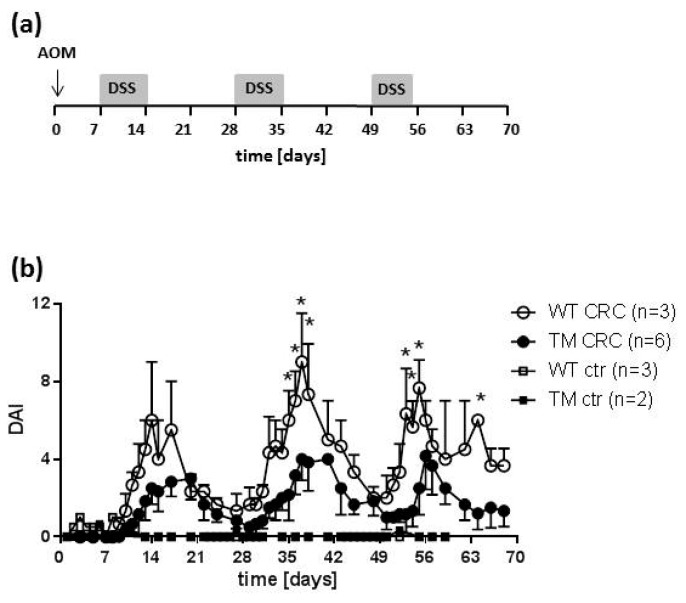
(**a**) Schematic representation of the schedule of acetoxymethan/dextran sulfate sodium (AOM/DSS) application. Control mice were treated with AOM only, but no DSS was applied (not shown). (**b**) Mice of either wild type (WT) or H_4_R^−/−^ (TM) genotype were treated either with AOM/DSS (CRC) or with AOM only (ctr). On the days indicated, clinical disease symptoms, body weight, perianal bleeding, and stool consistency, were monitored, graded by a scoring system, and summed up to the disease activity index (DAI). Means +/− SD of *n* = 2–6. *, *p* < 0.05 (two-way ANOVA with Sidak’s post hoc test).

**Figure 2 cancers-12-00912-f002:**
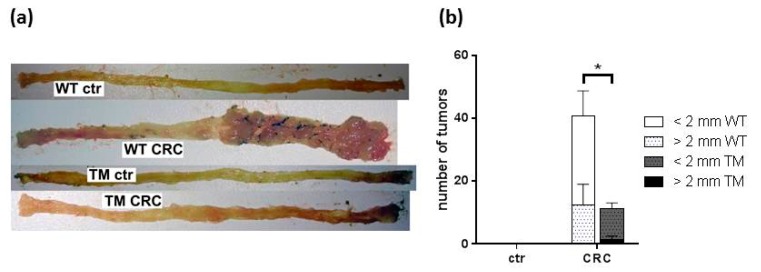
(**a**) Representative photographs of the colon specimen prepared out of the mice described in Figure 1. (**b**) The tumors detected in the colon specimen were counted and graded according to their diameter being either smaller or larger than 2 mm. WT: wild type, TM: H_4_R^−/−^, ctr: AOM-treated mice, CRC: AOM/DSS-treated mice. Means +/- SD of *n* = 2–6. *, *p* < 0.05 (two way ANOVA with Sidak’s post hoc test).

**Figure 3 cancers-12-00912-f003:**
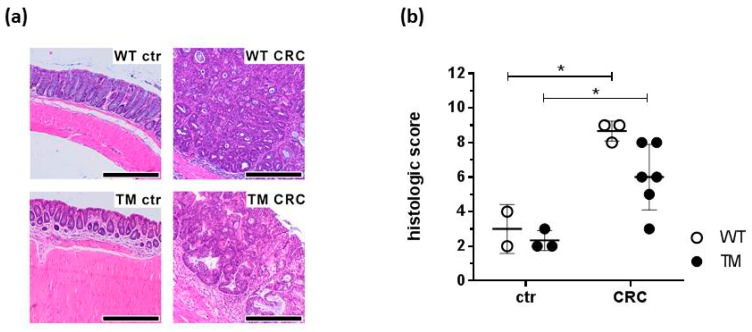
Colon tissues prepared out of the mice described in Figure 1 were fixed in buffered formalin, embedded in paraffin, and sliced into histological sections. Sections were stained with hematoxilin/eosin and evaluated for inflammation and tumor morphology. (**a**) Representative micro-photographs of the colon sections. (**b**) Quantitative grading of the tumors. WT: wild type, TM: H_4_R^−/−^, ctr: AOM-treated mice, CRC: AOM/DSS-treated mice. Magnification: 40×; scale bar: 300 µm. Individual values and means +/− SD of *n* = 2–6. *, *p* < 0.05 (two-way ANOVA with Sidak’s post hoc test).

**Figure 4 cancers-12-00912-f004:**
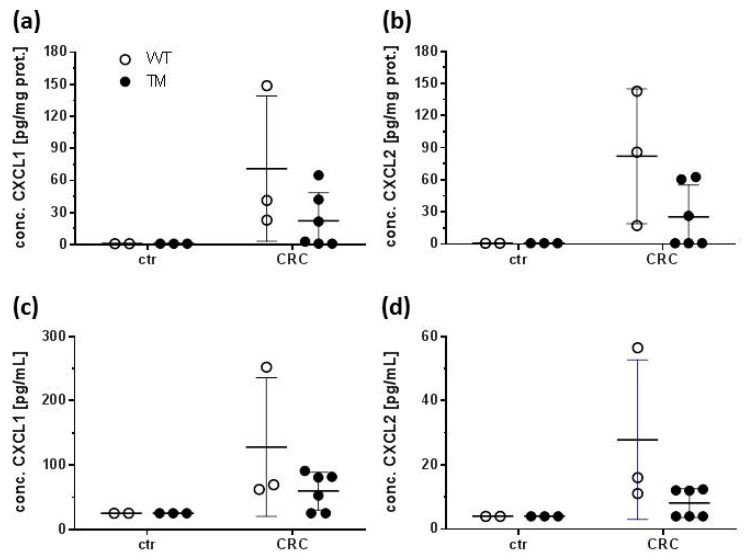
Colon homogenates and sera were prepared out of the mice described in Figure 1. The concentrations of CXCL1 and CXCL2 were quantified using a multiplex ELISA technique. (**a,b**) Concentrations of (**a**) CXCL1 and (**b**) CXCL2 in colon homogenates. (**c,d**) Concentrations of (**c**) CXCL1 and (**d**) CXCL2 in sera. WT: wild type, TM: H_4_R^−/−^, ctr: AOM-treated mice, CRC: AOM/DSS-treated mice. Individual values and means +/− SD of *n* = 2–6. No significant differences (two-way ANOVA with Sidak’s post hoc test).

**Figure 5 cancers-12-00912-f005:**
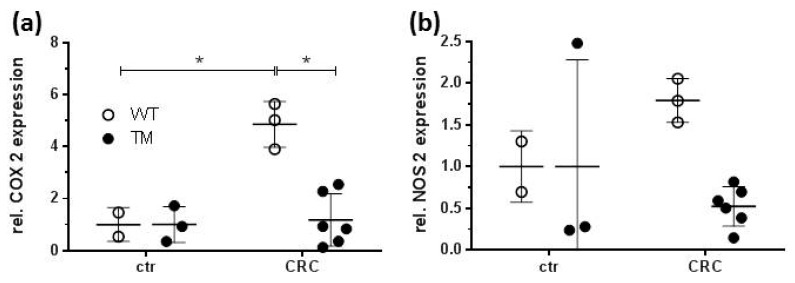
RNA was isolated from colon samples out of the mice described in Figure 1. The concentrations of mRNA encoding COX 2 and NOS 2 were quantified by RT-qPCR. (**a,b**) Relative mRNA expression of (**a**) COX2 and (**b**) NOS2. WT: wild type, TM: H_4_R^−/−^, ctr: AOM-treated mice, CRC: AOM/DSS-treated mice. Means +/− SD of *n* = 2–6. *, *p* < 0.05 (two-way ANOVA with Sidak’s post hoc test).

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
