# Peer review of "Genetic Deficiency of the Histamine H4-Receptor Reduces Experimental Colorectal Carcinogenesis in Mice"

_cancers, 2020, doi:10.3390/cancers12040912_

Round 1

Reviewer 1 Report

I think this is an interesting study showing that H4R involves in inflammation-associated colorectal carcinogenesis. Experiments were carefully conducted. However, several points listed below should be considered before publication.

1) Abstract section should be rewritten. Background is too long and the results are too short.

2) Figure 3: TM CAC must be a tubular adenoma, but not CAC. I’m pathologist and this is my opinion.

3) Immunohistochemistry of H4R in non-lesional colonic mucosa and CRC of WT and TM mice is quite interesting. Please provide data and/or photos.

4) Please explain the word “cola” in the text.

Author Response

I think this is an interesting study showing that H4R involves in inflammation-associated colorectal carcinogenesis. Experiments were carefully conducted. However, several points listed below should be considered before publication.

1) Abstract section should be rewritten. Background is too long and the results are too short.

       We thank the reviewer for this warrantable demur. We revised the abstract according to her/his comment.

2) Figure 3: TM CAC must be a tubular adenoma, but not CAC. I’m pathologist and this is my opinion.

       Thanks to the reviewer for indicating this point. Indeed, there was no case of infiltration of the tunica muscularis. We changed the mentioned panel in Figure 3 and edited the histology part in the revised version of the manuscript.

3) Immunohistochemistry of H4R in non-lesional colonic mucosa and CRC of WT and TM mice is quite interesting. Please provide data and/or photos.

        Unfortunately, we were not able to perform immunohistochemical analyses of H4R, since we still did not identify reliable antibodies recognizing H4R (Commercially available antibodies against human and murine histamine Hâ‚„-receptor lack specificity. Beermann S, Seifert R, Neumann D. Naunyn Schmiedebergs Arch Pharmacol. 2012, 385:125-35). We also tried in-situ hybridization (unpublished), however, without success. Thus, we cannot provide data on the spatial differential expression of H4R in tumor vs. non-tumor mucosal cells.

4) Please explain the word “cola” in the text.

        According to Merriam-Websters dictionary and others, the plural of ‘colon’ is ‘colons’ or ‘cola’. Here, we opted to use the term ‘cola’ as plural of ‘colon’.

Reviewer 2 Report

The aim of the work by Schirmer and coworkers is to address the role of Histamine and the H4R receptor in colitis-associated colon cancer (CAC). For this purpose, they used the AOM/DSS model in mice. The data show that H4R KO Mice are less sensitive to the AOM/DSS treatment than wild type mice. The results are interesting but mainly descriptive.

Comments:

1) Although the AOM/DSS model is widely described for CAC, this is absolutely not the case. Even the authors in their introduction describe cancer-associated colon cancers and sporadic cancers in such terms that AOM/DSS cannot be a relevant model of CAC. Indeed, Schirmer points that “inflammation rarely precedes sporadic or hereditary CRC” whereas in CAC the chronic inflammatory disease precedes cancer. In other words, in sporadic cancer the oncogenic event marks the onset of the disease whereas in CAC the disease starts with chronic inflammation and subsequently evolves into cancer. In the AOM/DSS model, the first event in not inflammation but the induction of oncogenic mutations by AOM, while DSS treatment to induce inflammation is the secondary event 1 week after AOM. Therefore, instead of a model of CAC, the AOM/DSS model is a model of chemical carcinogenesis amplified by the DSS treatment. The amplification effect of DSS may result from the increased epithelial cell proliferation needed to repair the surface of the injured mucosa as well as from the local inflammation linked to the invasion of the mucosa by the microbiote.

Thus, the aim and the conclusion of the manuscript must be reconsidered according to this remark.

2) The authors measured CXCL1 / CXCL2 levels as well as the mRNA expression of COX2 and NOS2 in the whole mucosa of the mice. What are the cell types responsible for these changes?

3) A better histological and immuno-histological description of the tumors with markers of tumor cells and immune cells is needed.

Author Response

Reviewer 2

The aim of the work by Schirmer and coworkers is to address the role of Histamine and the H4R receptor in colitis-associated colon cancer (CAC). For this purpose, they used the AOM/DSS model in mice. The data show that H4R KO Mice are less sensitive to the AOM/DSS treatment than wild type mice. The results are interesting but mainly descriptive.

Comments:

1) Although the AOM/DSS model is widely described for CAC, this is absolutely not the case. Even the authors in their introduction describe cancer-associated colon cancers and sporadic cancers in such terms that AOM/DSS cannot be a relevant model of CAC. Indeed, Schirmer points that “inflammation rarely precedes sporadic or hereditary CRC” whereas in CAC the chronic inflammatory disease precedes cancer. In other words, in sporadic cancer the oncogenic event marks the onset of the disease whereas in CAC the disease starts with chronic inflammation and subsequently evolves into cancer. In the AOM/DSS model, the first event is not inflammation but the induction of oncogenic mutations by AOM, while DSS treatment to induce inflammation is the secondary event 1 week after AOM. Therefore, instead of a model of CAC, the AOM/DSS model is a model of chemical carcinogenesis amplified by the DSS treatment. The amplification effect of DSS may result from the increased epithelial cell proliferation needed to repair the surface of the injured mucosa as well as from the local inflammation linked to the invasion of the mucosa by the microbiote.

Thus, the aim and the conclusion of the manuscript must be reconsidered according to this remark.

          We are grateful for the detailed comments from reviewer 2 and totally agree that the AOM/DSS model is a model of chemically induced carcinogenesis displaying some aspects of colorectal carcinoma (CRC). Indeed, due to the treatment schedule, it does not necessarily reflect colitis-associated CRC, as largely has been supposed. Therefore, we accordingly revised the text, incl. aim and conclusions.

2) The authors measured CXCL1 / CXCL2 levels as well as the mRNA expression of COX2 and NOS2 in the whole mucosa of the mice. What are the cell types responsible for these changes?

          We are sorry, that we have not unambiguously described this point in our manuscript. Regarding the effect of H4R, we favor the opinion that epithelial cells are responsible for the changes due to the lack of H4R expression. This is now explicitly pointed out in the revised version of our manuscript. We also added general information on CXCL1/CXCL2 expressing cells in the Discussion section of the revised version. COX 2 and NOS 2 are expressed rather ubiquitously, thus we do not comment on this. Regarding our data, NOS 2 is not regulated by H4R, thus, we do not discuss this point. 

3) A better histological and immuno-histological description of the tumors with markers of tumor cells and immune cells is needed.

           Thanks to the reviewer for this hint. We performed additional analyses and added such information in the Results section. As such, we performed immunohistological analyses of CD3- and CD22-expressing cells. The inflammatory tumor infiltrates consist of mostly CD3+ T cells, as expected; therefore, we added such notion as ‘data not shown’. Most of the antibodies used for immunohistology in our hands worked not very well in formalin-fixed tissues. Cryopreservation would have been the better choice, but unfortunately, in this study we performed formalin fixation only. Thus, due to time constrains for the revision process and for legal reasons, we are not able to provide a comprehensive immunohistological analysis.